# The Eyelid Angiosarcoma: A Systematic Review of Characteristics and Clinical Course

**DOI:** 10.3390/jcm11144204

**Published:** 2022-07-20

**Authors:** Gyudeok Hwang, Jeongah Shin, Ji-Young Lee, Kyung-Sun Na, Ji-Sun Paik, Hyeon Woo Yim, Suk-Woo Yang, Won-Kyung Cho

**Affiliations:** 1Department of Ophthalmology, Daejeon St. Mary’s Hospital, College of Medicine, The Catholic University of Korea, Seoul 06591, Korea; mehwang78@naver.com (G.H.); nole@catholic.ac.kr (J.S.); ram1020@naver.com (J.-Y.L.); 2Department of Ophthalmology, Yeouido St. Mary’s Hospital, College of Medicine, The Catholic University of Korea, Seoul 06591, Korea; githen@hanmail.net (K.-S.N.); rollipopp@hanmail.net (J.-S.P.); 3Department of Preventive Medicine, College of Medicine, The Catholic University of Korea, Seoul 06591, Korea; y1693@catholic.ac.kr; 4Department of Ophthalmology, Seoul St. Mary’s Hospital, College of Medicine, The Catholic University of Korea, Seoul 06591, Korea; yswoph@hanmail.net

**Keywords:** angiosarcoma, eyelid, prognosis, systematic review, treatment

## Abstract

A systematic search for eyelid angiosarcoma was performed from inception to December 2020 in Medline, EMBASE, and the Cochrane databases. Forty two eyelid angiosarcoma cases in 32 articles were analyzed. Eyelid angiosarcomas showed an incidence peak in the eighth decade of life, and was reported more frequently in Caucasian males. Eyelid angiosarcomas were associated with a mortality rate of 26.2%, a recurrence rate of 14.3%, and a cure rate of 45.2%. Four years event-free survival (EFS) rate was 36.0%, with median EFS of 36 months. Eyelid angiosarcomas with bilateral involvement or metastasis showed higher mortality and recurrence rates than unilateral eyelid invasion cases. In the prognosis analysis according to treatment modalities, the mortality and recurrence rates were the lowest in patients who underwent surgical excision. The 4-year EFS probability in a group with surgical excision was 60.6%, but in a group without surgical excision it was 30.3%. A total of 45.2% of the cases was misdiagnosed and 21.4% of the cases could not be correctly diagnosed with the first biopsy trial. The prognosis for eyelid angiosarcomas was better than that of angiosarcomas invading the face and scalp. Surgical excision was the most important treatment modality; thus, should be considered as the first treatment of choice.

## 1. Introduction

A diverse range of malignancies are known to invade the eyelid. Epidermal malignancies such as basal cell carcinoma and squamous cell carcinoma, melanocytic-origin tumors such as malignant melanoma, adnexal-origin tumors such as sebaceous carcinoma, and sarcomas such as angiosarcoma and Kaposi’s sarcoma occur in the eyelid [1].

The soft tissue sarcomas are extremely rare malignant tumors that constitute less than 1% of all cancers, and 2% of soft tissue sarcomas are angiosarcomas [2]. Angiosarcomas can occur in any soft-tissue structure or viscera, but they most commonly present as cutaneous disease in elderly Caucasian men, and these cutaneous angiosarcomas usually appear on the head and neck [3]. An angiosarcoma invading the face and scalp has a poor prognosis, with a 5-year survival rate of 12% [4]. Even a localized disease shows a median survival of 7 months [3].

The question arises as to whether an angiosarcoma that occurs on the eyelid shows a poor prognosis in the same way as the others? In this study, we performed a systematic review of case reports using survival analysis to assess the characteristics, incidence, treatment modalities, tendency to misdiagnosis, and the prognosis of the eyelid angiosarcoma for the first time.

## 2. Materials and Methods

### 2.1. Search Strategy and Study Selection

A systematic search from inception to December 2020 was performed in the Medline, EMBASE, and the Cochrane databases, to identify patients diagnosed with angiosarcomas invading the eyelid. To briefly explain the search strategy, ‘Angiosarcoma’ and ‘Eyelid’ were used as the search expressions in Medline’s Medical Subject Headings (MeSH), Embase Subject Headings (Emtree), and the Cochrane Medical Terms. The inclusion criteria were as follows: (1) cases of angiosarcomas involving the eyelid; (2) identifiable demographic information for age and sex; and (3) identifiable treatment modality. The exclusion criteria were as follows: (1) non-human cases; (2) presence of malignancies or diseases other than angiosarcomas; (3) cases of angiosarcomas that did not involve the eyelid; (4) missing identifiable demographic information for age and sex; (5) missing information on treatment modality; and (6) literature reviews or comments without a case report.

The first or the corresponding author was contacted by an e-mail for the cases where a clear follow-up period or the treatment outcome could not be confirmed in the paper.

### 2.2. Data Extraction

Two investigators (HGD and CWK) independently selected the potentially eligible studies. The disagreements in study selection were resolved by discussions involving the investigators. Finally, 32 articles [5,6,7,8,9,10,11,12,13,14,15,16,17,18,19,20,21,22,23,24,25,26,27,28,29,30,31,32,33,34,35,36] describing a total of 42 cases were selected (Figure 1).

The information regarding the year and country in which each case was published; age, sex, and ethnic group of the patients; site of the eyelid angiosarcoma; treatment modality, treatment outcome, and any misdiagnosed diseases; biopsy trials prior to pathological confirmation; and surgical reconstruction method were extracted from each published case.

### 2.3. Statistical Approach

The binomial exact test, survival analysis, and Cox regression were used for the statistical approach with 95% confidential interval. Statistical analyses were performed using the R program (R Foundation for Statistical Computing, Vienna, Austria), version 4.1.1 for Windows, and package survival for R. All of the procedures were performed according to the preferred reporting items for systematic reviews and meta-analyses (PRISMA) guideline, and the PRISMA 2020 checklist is shown in Appendix A).

### 2.4. Quality Assessment

For the 32 studies selected, a quality assessment was performed, using ‘a protocol for the use of case reports/studies and case series in systematic reviews’, developed by Nambiema et al. [37]. Two investigators (HGD and CWK) analyzed independently and resolved divergences through consensus.

## 3. Results

### 3.1. Characteristics of the Included Cases

The literature search yielded 42 eyelid angiosarcoma cases, and the findings for these cases are summarized in Table 1. Using ‘the critical appraisal checklist for case reports’, developed by the Joanna Briggs Institute, we assessed the included studies in our analysis, and the results are summarized in Appendix A).

### 3.2. Demographic and Laterality Characteristics of the Eyelid Angiosarcoma

The patient age in the eyelid angiosarcoma cases ranged from 5 months to 92 years. The mean (±standard deviation) age was 69.2 ± 18.2 years, and the median age was 75.5 years. The number of patients over 70 years of age was 27 (64.3%), and 14 patients (33.3%) were aged 70–79 years. The incidence peak was observed in the eighth decade of life (Appendix A).

When the cases were categorized by ethnic group, they were found to be most common among Caucasians (17 cases; 40.5%), followed by Asians (4 cases; 9.5%). The ethnic group could not be identified in 19 cases (Table 1). Among the 42 patients, 28 were men and 14 were women. The male:female ratio was 2:1, with a statistically significant male dominance in the binomial exact test (*p* = 0.012) (Appendix A).

Of the 42 cases, five (11.9%) involved both of the eyes, and 37 cases (88.1%) involved a single eye, with a statistically significant unilateral dominance in the binomial exact test (*p* < 0.001). Of the 37 cases involving the eyelids of a single eye, 10 (27.0%) involved the eyelids of the right eye, and 27 (73.0%) involved the eyelids of the left eye. The right:left eyelid ratio was 1:2.7, with a statistically significant left eyelid dominance in the binomial exact test (*p* = 0.003). A total of 18 cases (unilateral, 17 cases; bilateral, 1 case) involved only the upper or lower eyelids. Ten cases involved the upper eyelids, and eight cases involved the lower eyelids. Canthus was regarded as the upper and lower eyelid (Table 1). There was no statistically significant difference in the binomial exact test (*p* = 0.167) between the upper or lower eyelids.

### 3.3. Overall Prognosis of the Eyelid Angiosarcoma

Among the 42 patients, 11 (26.2%) died, 6 (14.3%) showed recurrence, and 19 (45.2%) were cured without recurrence (Figure 2; Table 2). In six cases [16,23,24,30,34,36], the treatment outcome could not be confirmed in the paper (Figure 2). These six cases were excluded in the survival analyses with 95% confidential interval (CI) (Figure 3 and Figure 4). In four cases [6,12,29,31], the follow-up period was not clearly described in the paper and ‘one month’ was used as a substituted value of the follow-up period in the survival analyses. The 4-year overall survival probability of patients with eyelid angiosarcomas was 48.7% (95% CI 25.0–94.9%), 3-year 64.9% (95% CI 45.6–92.4%), 2-year 81.2% (95% CI 68.5–96.1%), and 1-year 81.2% (95% CI 68.5–96.1%) (Appendix A). The median overall survival time was 48 months (Appendix A). If the event is defined as a recurrence and a death, the 4-year event free survival (EFS) probability of patients with an eyelid angiosarcoma was 36.0% (95% CI 17.6–73.6%), 3-year 48.0% (95% CI 30.9–74.4%), 2-year 65.4% (95% CI 50.5–84.7%), and 1-year 69.8% (95% CI 55.6–87.5%) (Figure 3). The median EFS time was 36 months (Figure 3).

### 3.4. Prognosis of the Eyelid Angiosarcoma According to Age and Laterality

In the analyses based on patient age, among the six patients aged under 50 years, one (16.7%) died, three (50.0%) showed recurrence, and two (33.3%) showed no recurrence. Among the nine patients aged between 50 and 70 years, two (22.2%) died, one (11.1%) showed recurrence, and five (55.6%) showed no recurrence (Table 1). Among the 27 patients aged over 70 years, 8 (29.6%) died, 2 (7.4%) showed recurrence, and 12 (44.4%) showed no recurrence (Table 1).

Among the five patients showing bilateral involvement, three (60.0%) died, one (20.0%) showed recurrence, and one (20.0%) showed no recurrence (Table 1). The eyelid angiosarcoma with bilateral involvement showed higher mortality and recurrence rates than the other cases.

### 3.5. Prognosis of the Eyelid Angiosarcoma According to Treatment Modalities

The treatment modalities used for eyelid angiosarcomas included surgical excision (including anterior orbitotomy and exenteration), radiation therapy, and chemotherapy. Antibiotics, steroids, and antihistamines were used for other misdiagnoses, such as cellulitis or angioedema, and complications, such as superimposed infection. Two patients [19,23] refused treatment for the eyelid angiosarcoma and one of them [19] showed spontaneous regression without recurrence for three years. We carefully suspect that this spontaneous regression case was misdiagnosed as an angiosarcoma.

Among the 40 patients who received treatment for angiosarcomas, 16 (40.0%) were treated with monotherapy using only surgical excision, radiation therapy, or chemotherapy, and 24 (60.0%) were treated with combination therapy. The combination therapy was used more frequently. The combination of surgical excision and radiation therapy was the most common (11 cases; 27.5%) (Table 2). Among the 14 patients who received treatment without surgical excision, such as radiation therapy alone or chemotherapy alone or a combination of radiation and chemotherapy, seven (50.0%) died and three (21.4%) showed recurrence, and only two (14.3%) patients did not show recurrence. On the other hand, among the 26 patients who underwent surgical excision, 4 (15.4%) died, 3 (11.5%) showed recurrence, and 16 (61.5%) showed no recurrence (Table 2). The comparison of EFS in the patients who underwent and did not undergo surgical excision is shown in Figure 4. The 4-year EFS probability in a group who underwent surgical excision was 60.6% (95% CI 26.9–100.0%), 3-year 90.9% (95% CI 79.7–100.0%), 2-year 90.9% (95% CI 79.7–100.0%), 1-year 90.9% (95% CI 79.7–100.0%) (Figure 4). On the other hand, the 4-year EFS probability in a group who did not undergo surgical excision was 30.3% (95% CI 10.0–91.6%), 3-year 30.3% (95% CI 10.0–91.6%), 2-year 60.6% (95% CI 36.2–100.0%), 1-year 60.6% (95% CI 36.2–100.0%) (Figure 4). The median EFS of the patients who underwent and did not undergo surgical excision were 48 and 12 months, respectively (Figure 4). If the log-rank test is used for comparison, the EFS distributions of two groups, the surgical excision significantly decreased the mortality and recurrence rate of patients with eyelid angiosarcomas (*p* = 0.015) (Figure 4).

### 3.6. Prognosis of the Eyelid Angiosarcoma According to Metastasis

Of the 42 cases, CT was used as a systemic workup in 16 cases [8,11,12,15,18,21,23,24,25,26,29,31,34,35,36], and MRI in 6 cases [9,11,18,21,26,31]. Only two cases [26,30] used PET-CT as a systemic workup.

Five patients showed metastasis to other organs, and all of them died. The metastasized organs varied, including the parotid gland, neck, bone, lung, and liver (Table 1). Recurrence or death occurred within 5 years in all of the cases. In four (66.7%) of the six recurrence cases, the recurrence occurred in less than 1 year. In 6 of the 11 death cases (54.5%), death occurred in less than 1 year (Table 1; Figure 2).

### 3.7. Misdiagnosis of the Eyelid Angiosarcoma

Of the 42 cases, 19 (45.2%) were misdiagnosed before the pathologic confirmation of an eyelid angiosarcoma. Cellulitis was the most common incorrect diagnosis (eight cases), followed by angioedema (five cases), and hematoma (three cases) (Table 1). Even when infection was confirmed in culture, the treatments for the infection did not show any improvement; finally, eyelid angiosarcoma was diagnosed in one case [34]. In two cases [23,31], hematoma and edema after periorbital contusion did not improve and eyelid angiosarcoma was diagnosed. Another case [11] showed abnormal findings in the thyroid function test but was diagnosed as eyelid angiosarcoma. The 42 cases included nine cases (21.4%) in which angiosarcoma was not diagnosed after the first biopsy trial and one case [34] where angiosarcoma was diagnosed only after three biopsy trials (Table 1). After the lesion of the eyelid angiosarcoma was surgically removed, various reconstruction methods, such as forehead flap or free skin graft, were used (Table 1).

### 3.8. Presenting Features of the Eyelid Angiosarcoma

The presenting features of the eyelid angiosarcoma were diverse.

Looking at the color features first, reddish/erythematous/rash-like changes were the most common color features in 18 cases, followed by bluish/violaceous/ecchymosis in 8 cases. However, there were two cases of a yellowish or pale yellow. There were some cases where the color features were not mentioned.

The shape features of the eyelid angiosarcoma were also presented in various ways. Mass-like lesions, such as nodule/papule/plaque, were presented in 19 cases. Swelling or edema features were mentioned in 18 cases. In addition, there were cases with macular lesions or only color-changed features.

The most common feature of the eyelid angiosarcoma can be briefly summarized as a reddish or bluish maculopapular lesion. However, it is necessary to be careful, as there are cases where it appears only with a change of color, or only the shape changes without a change of color.

### 3.9. Quality Assessment

The quality of the included studies was assessed using ‘a protocol for the use of case reports/studies and case series in systematic reviews’, summarized in Figure 5 and Figure 6.

## 4. Discussion

### 4.1. Incidence of the Eyelid Angiosarcoma

The incidence of angiosarcomas of the face and scalp was reported to peak in the eighth decade [4]; the eyelid angiosarcomas, similar to the face and scalp angiosarcomas, also showed a peak incidence in the eighth decade. Thus, the incidence of the eyelid angiosarcomas increased with age, but decreased after 80 years of age. We believe that the lower incidence after 80 years may be attributed to the smaller population over 80 years of age (Appendix A).

Angiosarcoma most commonly presents as a cutaneous disease in elderly Caucasians (Table 1) [3]. The most common malignant eyelid tumors are basal cell carcinomas in Caucasians and sebaceous gland carcinomas in Asians [38]. These differences may be attributable to genetic factors or differences in the environmental factors. The angiosarcomas of the face and scalp were reported to be prevalent in males [4]. Eyelid angiosarcomas also showed a higher incidence in males (Appendix A).

### 4.2. Predominance and Metastasis of the Eyelid Angiosarcoma

The eyelid angiosarcomas showed unilateral and left-eye predominance in the cases assessed in this review (Table 1). However, bilateral involvement was also observed in five cases (11.9%), indicating the need for thorough evaluation and follow-up of the contralateral eye in the cases of a unilateral eyelid angiosarcoma. It is difficult to accurately explain why eyelid angiosarcomas are frequent in the left eye, and data from more cases will be required to evaluate the left predominance.

There was no predominance in the location of the eyelid angiosarcoma between the upper and lower eyelids (Table 1). Lymphatic drainage of most of the upper eyelid and the lateral half of the lower eyelid is known to be directed to the preauricular nodes, while the medial portion of the upper and lower eyelids drains to the submandibular nodes; these two lymph nodes then connect to the nodes lower down the neck and drain through the deep cervical nodes. In 2010, Nijhawan et al. [39] performed lymphoscintigraphy on the eyelids of 28 patients and reported that the preauricular lymph nodes were the most common sentinel lymph nodes, regardless of the site of injection. Therefore, in the patients showing an eyelid angiosarcoma, an evaluation of the cervical lymph nodes, especially preauricular lymph nodes, through imaging studies, such as facial or neck CT, should be performed considering lymphatic metastasis.

Eyelid angiosarcomas metastasizing to other organs have a poor prognosis. Among the 42 patients, 5 showed metastasis to other organs, and all 5 patients died (Table 1). Since metastasis may occur not only in the parotid gland and neck, but also in the bone, lung, and liver, PET-CT should be performed to ensure that there is no metastasis in the whole body. The mortality and recurrence rates were both higher when the eyelid angiosarcoma involved both eyes than in cases involving a single eye. This is similar to the finding that survival has an important correlation with the extent of the primary tumor in cases of cutaneous angiosarcomas [40].

In cases showing recurrence or death, more than half of the cases showed these events in less than one year (Figure 2). Thus, short-term follow-up is recommended within the first year. One case [21] showed a recurrence at 29 months. Therefore, we recommend follow-up assessments for at least 5 years, similar to other cancers.

### 4.3. Prognosis of the Eyelid Angiosarcoma

Holden et al. [4] reported a 5-year survival rate of 12% in 72 patients with angiosarcomas invading the face and scalp, and found that 50% of the patients died within 15 months. Pawlik and colleagues [41] reported that the recurrence rate was 72% in 29 patients with angiosarcomas of the scalp. On the other hand, the mortality rate of the eyelid angiosarcomas was 26.2%, and the recurrence rate was 14.3%. The cure rate without recurrence in patients with an eyelid angiosarcoma was 45.2% (Table 2; Figure 2). Four years overall survival probability and 4-year EFS probability of the patients with an eyelid angiosarcoma were 48.7% (95% CI 25.0–94.9%) and 36.0% (95% CI 17.6–73.6%), respectively (Figure 3; Appendix A). These data show that the prognosis for eyelid angiosarcomas is better than that for angiosarcomas invading the face and scalp. This could be attributable to the fact that the eyelid angiosarcomas are detectable earlier than scalp lesions, allowing earlier interventions and reducing the mortality and recurrence rates. The mortality rate of the patients with eyelid angiosarcomas increased with age, but it was not clear whether this was due to age- or disease-related causes (Table 1).

### 4.4. Treatments of the Eyelid Angiosarcoma

Mendenhall et al. [40] reported that the optimal treatment for cutaneous angiosarcomas is the surgical removal of the lesion with a wide negative margin and postoperative radiotherapy to the primary site and regional lymphatics. The most commonly used treatment for eyelid angiosarcomas was combination therapy, with surgical excision and radiotherapy. When surgical excision was used as a treatment modality in eyelid angiosarcomas, the mortality rate decreased 3.3-fold, and the recurrence rate decreased 1.9-fold in comparison with the corresponding rates for the treatments without surgical excision (Table 2). The median EFS of the patients who underwent and did not undergo surgical excision were 48 and 12 months, respectively (Figure 4). In addition, the mortality and recurrence rate of patients treated with the surgical excision were significantly low (*p* = 0.015) (Figure 4). Thus, surgical excision is important to reduce the mortality rate and increase the cure rate.

Of the 42 cases, there were 26 cases in which excision was performed (Table 2). Mohs excision was performed in one case [22], exenteration in two cases [11,31], and anterior orbitotomy in one case [31]. There were 13 cases that clearly stated that a wide excision was performed. The treatment modality used in the other 10 cases was mentioned as surgical excision, complete surgical excision, En Bloc excision, or radical excision. However, most of them seem to have implemented wide excision. In addition, an intraoperative frozen margin was confirmed only in four cases [22,27,33,36]. Therefore, in most of the cases, a wide excision was performed, and the number of surgical cases was only 26, making it difficult to evaluate the difference in prognosis according to the surgical methods.

Although the prognosis of the eyelid angiosarcoma is better than that of angiosarcomas invading the face and scalp, the mortality rate is still high in cases of recurrence or metastasis. Mohs excision can be attempted only when a wide excision is not possible. In addition, it seems necessary to confirm whether the intraoperative frozen margin is negative even in these cases.

### 4.5. Misdiagnosis of the Eyelid Angiosarcoma

Eyelid angiosarcomas were misdiagnosed as another disease in 19 cases (45.2%), representing almost half of the 42 cases (Table 1). In one case [34], a cellulitis lesion with infection confirmed in culture was finally diagnosed as an eyelid angiosarcoma after three biopsy trials. In another case [24], an eyelid angiosarcoma was diagnosed after repeat biopsies for a hematoma that occurred after a periorbital contusion. Thus, an eyelid angiosarcoma should be considered in the differential diagnosis of eyelid lesions that appear to be cellulitis, angioedema, or hematoma but do not respond to the respective treatments. In such cases, the correct diagnosis may require several biopsy trials.

The eyelid angiosarcoma is prone to misdiagnosis. If treatments for other misdiagnosed eyelid problems, such as cellulitis or angioedema, are not effective, eyelid angiosarcoma should be suspected and confirmed through repeated biopsies.

### 4.6. Limitation and Strengths of this Study

A limitation of our analysis is related to selection bias. The angiosarcoma is rare and the included articles in this study are case reports and case series, which they are inherently associated with selection bias: publication or reporting bias. However, Sampayo-Cordero et al. [42] demonstrated that the systematic reviews of case reports are necessary to fully summarize the state of research in rare diseases.

To the best of our knowledge, this is the first systematic review of eyelid angiosarcomas. The quantitative synthesis of prognosis was effectively performed, using survival analysis from case reports and case series.

### 4.7. Conclusions and Future Research

In conclusion, the prognosis for eyelid angiosarcomas was better than that of angiosarcomas invading the face and scalp. Surgical excision is the most important treatment for reducing the mortality and recurrence rates of eyelid angiosarcomas. Therefore, if there is no clear evidence of distal metastasis, surgical resection should be performed by an ophthalmologist. 

In the future, if more cases of eyelid angiosarcomas are reported, the research on the most effective method among the various surgical excision methods should be conducted. In addition, prospective studies, such as randomized clinical trials (RCTs), can be designed to determine the prognosis difference between radiotherapy and chemotherapy with surgical excision.

## Figures and Tables

**Figure 1 jcm-11-04204-f001:**
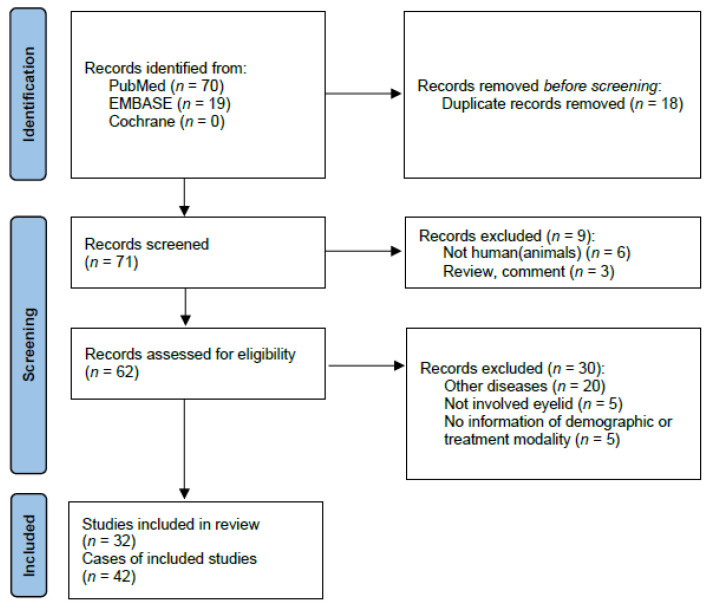
Flow diagram of study collection through exclusion criteria in this review.

**Figure 2 jcm-11-04204-f002:**
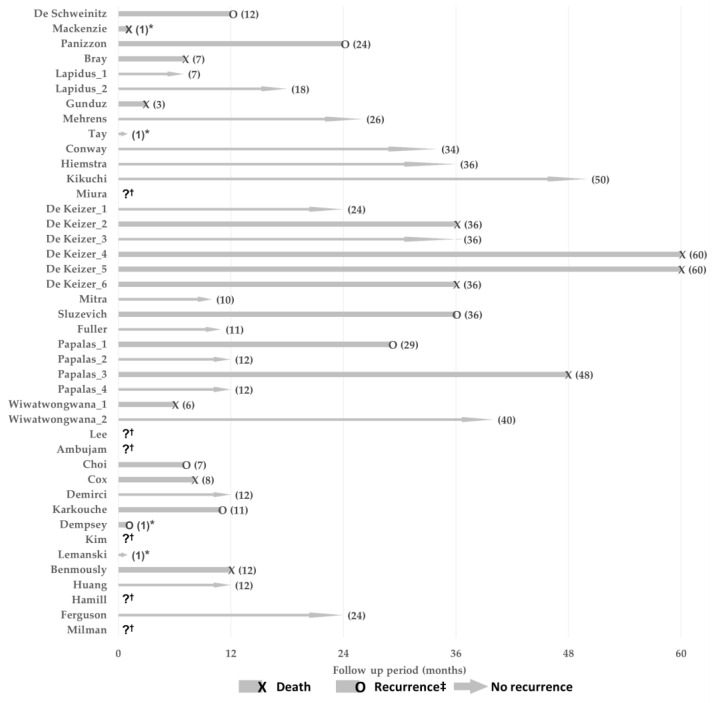
Follow-up period and treatment outcome of each case [5,6,7,8,9,10,11,12,13,14,15,16,17,18,19,20,21,22,23,24,25,26,27,28,29,30,31,32,33,34,35,36]. * Cases which clear follow-up period could not be confirmed in the paper were indicated as 1 month, † Cases which the treatment outcome could not be confirmed in the paper were left blank with a question mark, ‡ Recurrence with disease worsening.

**Figure 3 jcm-11-04204-f003:**
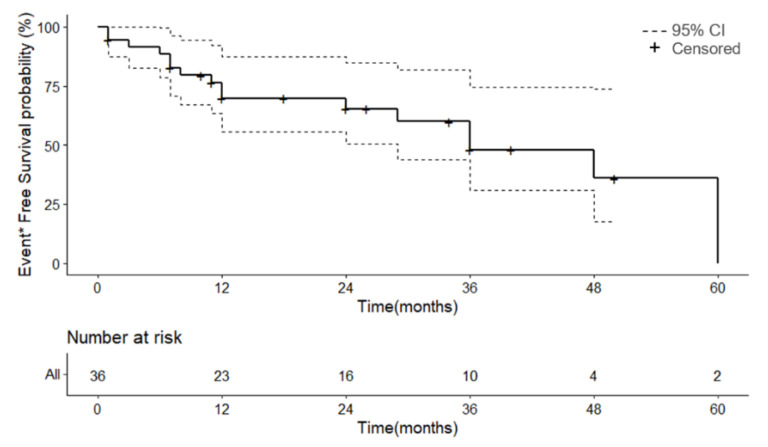
Event-free survival probability of patients with eyelid angiosarcoma. CI = confidence interval, * Event includes recurrence and death.

**Figure 4 jcm-11-04204-f004:**
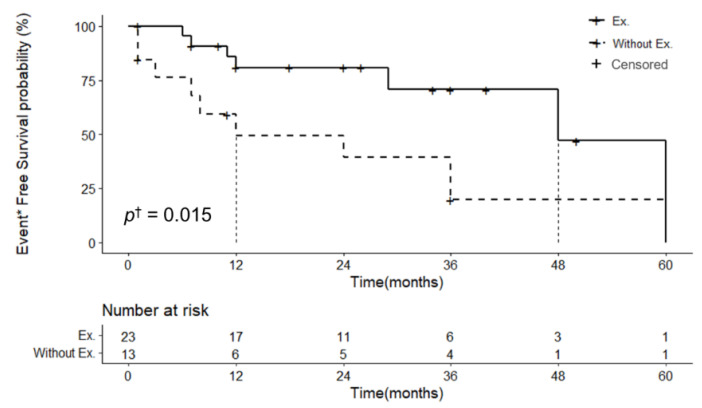
Comparison of event-free survival in patients who underwent and did not undergo surgical excision. Ex. = Surgical excision, * Event includes recurrence and death, ^†^ Log-rank test was used for comparison the survival distributions of two groups.

**Figure 5 jcm-11-04204-f005:**
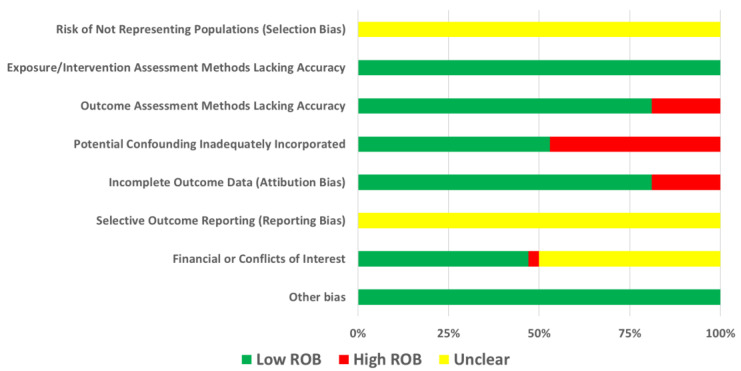
Risk of bias graph. ROB = Risk of bias.

**Figure 6 jcm-11-04204-f006:**
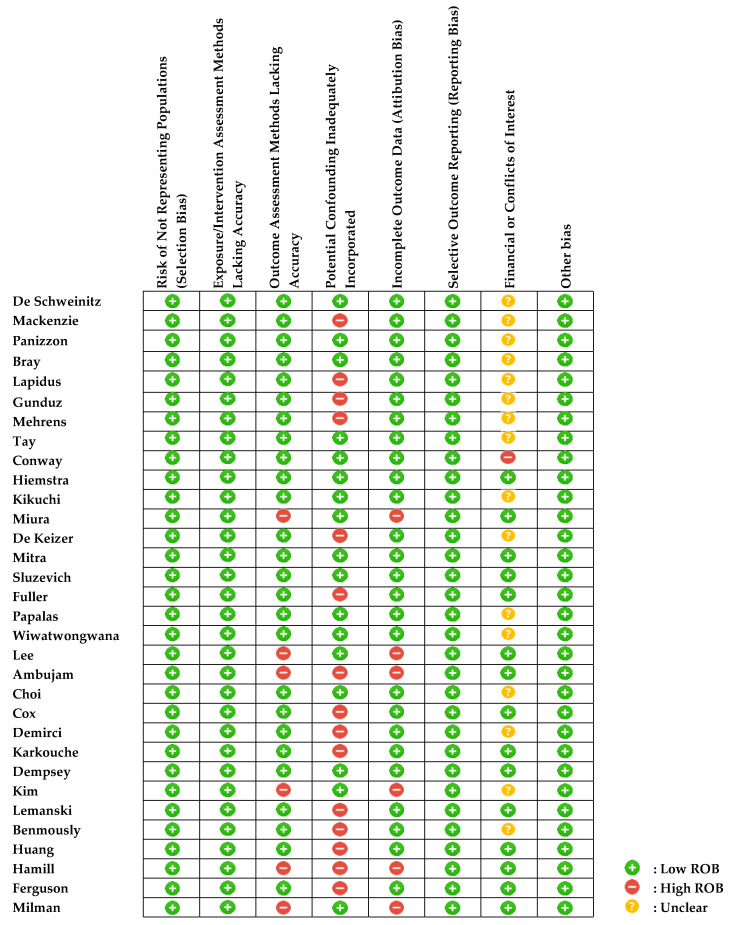
Risk of bias summary: judgements about each risk of bias item for each included study [5,6,7,8,9,10,11,12,13,14,15,16,17,18,19,20,21,22,23,24,25,26,27,28,29,30,31,32,33,34,35,36]. ROB = Risk of bias.

**Table 1 jcm-11-04204-t001:** Summary of the findings in the 42 published cases of eyelid angiosarcoma.

First Author	Year	Published Country	Ethnic Group of the Patient	Age/Sex	Involved Site	Mis-Diagnosis	Biopsy Trials	Surgical Recon.	Metastasis	Treatment Modality	Treatment Outcome
De Schweinitz [5]	1920	USA	Caucasian	5 mo/F	RLL	N	1	DS	N	Ex, RT	Recurred in 1 yr.
Mackenzie [6]	1985	UK	N/A	67/M	BUL, BLL	Cel., AE	2	FF, ERF	Bone	RT, Steroids	Died of metastasis
Panizzon [7]	1990	CHE	Caucasian	84/M	RLL	Cel.	1	N	N	EBRT	Recurred in 2 yr.
Bray [8]	1995	AU	Caucasian	79/F	LUL	N	1	N	N	Ex	Died of MI in 7 mo.
Lapidus_1 [9]	1996	USA	N/A	82/M	LUL, LMC	N	1	N	N	Ex, EBRT	No Recur. for 7 mo.
Lapidus_2 [9]	1996	USA	N/A	65/M	RUL, RMC	N	1	N	N	Ex, EBRT	No Recur. for 18 mo.
Gunduz [10]	1998	USA	N/A	82/M	LUL, LLL	N	1	N	PG, Ribs	EBRT	Died of metastasis in 3 mo.
Mehrens [11]	2000	DEU	N/A	48/F	LUL, LLL	TED	1	SSG	N	Exenteration, RT, Steroids	No Recur. for 26 mo.
Tay [12]	2000	SGP	Asian (CHN)	54/M	BUL, BLL	AE	1	N	N	RT, Chemo, Steroids	No Recur.
Conway [13]	2003	DEU	Caucasian	69/M	RUL, RLL, RMC	N	1	N	N	Ex, EBRT	No Recur. for 34 mo.
Hiemstra [14]	2004	NLD	Caucasian	71/M	LUL, LMC	Cel.	2	FF, TSF, FSG	N	Ex, Anti.	No Recur. for 3 yr.
Kikuchi [15]	2004	JPN	Asian (JPN)	75/M	RUL	N	1	N	N	Ex, RT, IL-2	No Recur. for 50 mo.
Miura [16]	2006	JPN	Asian (JPN)	92/M	RLL	N	1	N	N	Ex, RT, Chemo, rIL-2	N/A
De Keizer_1 [17]	2008	NLD	N/A	80/F	Left eyelid	N	1	N	N	Ex	No Recur. for 2 yr.
De Keizer_2 [17]	2008	NLD	N/A	75/F	Left eyelid	N	1	N	Neck, LN	Chemo	Died of metastasis in 3 yr.
De Keizer_3 [17]	2008	NLD	N/A	60/M	LLL	Cel.	2	N	N	Ex, RT	No Recur. for 3 yr.
De Keizer_4 [17]	2008	NLD	N/A	86/M	BUL	N	1	N	N	Chemo	Died in 5 yr.
De Keizer_5 [17]	2008	NLD	Caucasian	87/M	LUL, LLL	AE	2	N	N	Ex, Chemo, Steroids	Died in 5 yr.
De Keizer_6 [17]	2008	NLD	N/A	46/F	RUL, RLL	N	1	N	N	Chemo	Died in 3 yr.
Mitra [18]	2008	UK	Caucasian	43/F	LLL	N	1	FSG	N	Ex, RT	No Recur. for 10 mo.
Sluzevich [19]	2008	USA	Caucasian	92/M	LUL	N	1	N	N	Observation	No Recur. for 3 yr. after spontaneous reg
Fuller [20]	2010	USA	N/A	77/M	LUL, LLL	N	1	N	N	Chemo	No Recur. for 11 mo.
Papalas_1 [21]	2010	USA	Caucasian	56/F	LLL, LMC	N	1	PF	N	Ex, EBRT, Avastin	Recur. in 29 mo.
Papalas_2 [21]	2010	USA	Caucasian	80/M	LUL	Cha.	1	N	N	Ex, Chemo	No Recur. for 1 yr.
Papalas_3 [21]	2010	USA	Caucasian	73/M	LLL	N	1	N	N	Ex, Chemo	Died in 4 yr.
Papalas_4 [21]	2010	USA	Caucasian	79/M	LLL	N	1	N	N	Ex, RT, Chemo	No Recur. for 1 yr.
Wiwatwongwana_1 [22]	2010	CAN	Caucasian	82/F	LUL	N	1	CRF, FF, FSG	Lung, Liver	Ex, RT	Died of metastasis in 6 mo.
Wiwatwongwana_2 [22]	2010	CAN	Caucasian	71/M	LUL	N	1	MTF	N	Ex, RT, Chemo	No Recur. for 40 mo.
Lee [23]	2012	USA	Hispanic	83/M	LUL, LLL	PH	1	N	N	Anti., Steroids	N/A
Ambujam [24]	2013	IND	N/A	65/M	LUL, LLL	PH	2	N	N	RT	N/A
Choi [25]	2013	USA	PI (SAM)	43/M	BUL, BLL	AE	2	N	N	Chemo, Antihistamine, Anti., Steroids	Worsening in 7 mo.
Cox [26]	2013	USA	N/A	76/M	LUL	Cel.	1	N	PG, Lung	Anti., Chemo, RT	Died of metastasis in 8 mo.
Demirci [27]	2013	USA	N/A	77/F	LLL	N	1	N	N	Ex	No Recur. for 1 yr.
Karkouche [28]	2013	FRA	N/A	21/F	LMC	N	1	N	N	Ex, Chemo	Recur. in 11 mo.
Dempsey [29]	2014	USA	Caucasian	82/M	LUL, LLL	Cel.	1	N	N	Chemo	Recurred shortly after Chemo
Kim [30]	2014	KOR	Asian (KOR)	71/M	RUL, RLL	N	1	N	N	RT, Chemo	N/A
Lemanski [31]	2014	USA	N/A	83/F	LUL, LLL	PH	1	SSG	N	Exenteration, RT, Ant. Orbitotomy	No Recur.
Benmously [32]	2016	TUN	N/A	64/M	BUL, BLL	N	1	N	N	RT, Chemo	Died in 1 yr.
Huang [33]	2015	CHN	N/A	76/F	LUL, LLL, LMC	N	1	FSG	N	Ex	No Recur. for 1 yr
Hamill [34]	2016	USA	N/A	77/M	RUL	Cel.	3	N	N	Ex, Anti.	N/A
Ferguson [35]	2018	USA	N/A	57/F	LUL, LLL	Cel., AE	2	N	N	Ex, Chemo, Anti.	No Recur. for 2 yr.
Milman [36]	2018	USA	Caucasian	76/M	RUL	HE	2	FF	N	Ex, RT, Chemo	N/A

Recon. = Reconstruction; USA = United States of America; UK = United Kingdom; CHE = Switzerland; AU = Australia; DEU = Germany; SGP = Singapore; NLD = Netherlands; JPN = Japan; CAN = Canada; IND = India; FRA = France; KOR = Korea; TUN = Tunisia; CHN = China; PI = Pacific Islander; SAM = Samoan; M = Male; F = Female; BUL = Both upper eyelids; BLL = Both lower eyelids; RUL = Right upper eyelid; RLL = Right lower eyelid; LUL = Left upper eyelid; LLL = Left lower eyelid; RMC = Right medial canthus; LMC = Left medial canthus; N = None; Cel. = Cellulitis; AE = Angioedema; TED = Thyroid eye disease; Cha. = Chalazion; HE = hemangioendothelioma; PH = Periorbital hematoma; DS = Direct Suture; FF = Forehead Flap; ERF = Eyelid Rotation Flap; SSG = Split Skin Graft; TSF = Tarsal Sliding Flap; FSG = Free Skin Graft; PF = Pedicle Flap; CRF = Cheek Rotation Flap; MTF = Modified Tenzel Flap; PG = Parotid gland; LN = Lymph node; Ex = Surgical excision; RT = Radiotherapy; EBRT = External beam radiotherapy; Chemo = Chemotherapy; Anti. = Antibiotics; rIL = Recombinant interleukin; Ant. = Anterior; MI = Myocardial infarction; mo. = Month(s); yr. = Year(s); N/A = Not available; Recur. = Recurrence; reg = Regression.

**Table 2 jcm-11-04204-t002:** Treatment outcomes according to modalities.

	Treatment Outcome, No. (%)				
Treatment Modality	No Recurrence	Recurrence ^‡^	Death	N/A	Total
Monotherapy					
Ex * only	4 (9.5)	-	1 (2.4)	1 (2.4)	6 (14.3)
RT ^†^ only	-	1 (2.4)	2 (4.8)	1 (2.4)	4 (9.5)
Chemo only	1 (2.4)	2 (4.8)	3 (7.1)	-	6 (14.3)
Combination Therapy					
Ex and RT	8 (19.0)	2 (4.8)	1 (2.4)	1 (2.4)	12 (28.6)
Ex and Chemo	2 (4.8)	1 (2.4)	2 (4.8)	-	5 (11.9)
RT and Chemo	1 (2.4)	-	2 (4.8)	1 (2.4)	4 (9.5)
Ex and RT and Chemo	2 (4.8)	-	-	1 (2.4)	3 (7.1)
Others ^§^	1 (2.4)	-	-	1 (2.4)	2 (4.8)
Total	19 (45.2)	6 (14.3)	11 (26.2)	6 (14.3)	42 (100.0)

No. = Number; Ex = Surgical excision; RT = Radiotherapy; Chemo = Chemotherapy; N/A = Not available. * Ex including exenteration; ^†^ RT including external beam radiotherapy; ^‡^ Recurrence including disease worsening; ^§^ A case treated by therapies other than Ex/RT/Chemo; and one observational case are classified as ‘others’.

## Data Availability

All data generated or analyzed during this study are included in this published article.

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
