# Peer review of "The Eyelid Angiosarcoma: A Systematic Review of Characteristics and Clinical Course"

_jcm, 2022, doi:10.3390/jcm11144204_

Round 1

Reviewer 1 Report

The authors performed a systematic review of the characteristics and clinical course of the eyelid angiosarcoma, and they found that prognosis of eyelid angiosarcomas was better that of angiosarcoma invading the face and scalp. Surgical excision was the most important treatment modality, should be considered as the first treatment choice.

The topic is of interest to plastic and reconstructive ophthalmologist surgeons. The systematic review procedure was performed with an excellent way, following the PRISMA recommendation.

Title should be recorded to give prior importance to the eyelid angiosarcoma as first words and then the rest. “of literature” is redundant

In abstract, please include additional information about the results, and in case of the limit word reach, reduce on the general first information.

The introduction is short and OK for the interest readers that could be plastic and reconstructive ophthalmic surgeons. At the end of the introduction, the purpose should be clearly stated and defined with more parameters that were measured on the results section.

Did you find interesting publication prior to 2000? Or maybe other year? In my opinion all systematic reviews should have a window search time to increase the comparability results between all studies. Incredibly old studies, probably have different methodology and advanced to the newest one.

If the flowchart was presented in a figure, please do not describe in word on the text again. Explain only one time.

Increase the visual appearance of the manuscript within the size of the items such tables and figure. One page could no be longer to have only the flowchart. Please revise.

The statistical approach should be explained in a separate section on methodology

Risk of bias is mandatory in the systematic review, include on the methodology section and results.

Update the PRISMA checklist to the latest version of 2020 and change the manuscript according to this updated issue.

The results were clearly stated and described but you need to divide into section to improve the readability. It is impossible to read all this text in one time without divisions.

Improve the visual appearance of the flowchart, use programs such lucid chart to better visuality. Reduce the size. Revise the sum of the articles.

Try to include table 1 in two pages, When the window time was applied it is probably eased to do this.

The discussion was well-written but needs section too. A strengths and limitations section should be added.

Furthermore, a future research lines could be added at the end of the discussion

The conclusion could be more soundness in order to give extra power to the research.

Author Response

Responses to the Reviewers

Reviewer 1

Comments and Suggestions for Authors

The authors performed a systematic review of the characteristics and clinical course of the eyelid angiosarcoma, and they found that prognosis of eyelid angiosarcomas was better that of angiosarcoma invading the face and scalp. Surgical excision was the most important treatment modality, should be considered as the first treatment choice.

The topic is of interest to plastic and reconstructive ophthalmologist surgeons. The systematic review procedure was performed with an excellent way, following the PRISMA recommendation.

à reply: Thank you very much for the reviewer's very detailed and kind review and comments on this humble study.

Title should be recorded to give prior importance to the eyelid angiosarcoma as first words and then the rest. “of literature” is redundant

à reply: Thank you for your suggestion. The title has been modified as follows. Line 2~3.

In abstract, please include additional information about the results, and in case of the limit word reach, reduce on the general first information.

à reply: In the abstract, additional information about the results especially prognosis according to the treatment modalities were included. Line 18~31. It looks more informative. Thank you for your comment.

Previous sentence of “The characteristics and clinical course of the eyelid angiosarcoma was evaluated by a systematic review of literature” was removed. Previous sentence of “Binomial exact test, survival analysis and Cox regression were used for the statistical approach with 95% confidential interval.” was moved to ‘2.3. Statistical approach’ section. Previous sentence of “And unilateral invasion was dominant; predominance in the upper or lower eyelids was not observed.” was removed.

The introduction is short and OK for the interest readers that could be plastic and reconstructive ophthalmic surgeons. At the end of the introduction, the purpose should be clearly stated and defined with more parameters that were measured on the results section.

à reply: At the end of the introduction, the purpose of this study was clarified by mentioning statistical analysis and results parameters. Line 47-49. The purpose looks delivered more clearly. Thank you for your comment.

Did you find interesting publication prior to 2000? Or maybe other year? In my opinion all systematic reviews should have a window search time to increase the comparability results between all studies. Incredibly old studies, probably have different methodology and advanced to the newest one.

à reply: Thank you for your inquiry. Our search window was from the inception (1920, reference [5]) to December 2020 and this search time window was described in abstract and line 52. Prior to 2000, six studies (reference [5~10]) and 7 cases were found. The first case in 1920 was a five-month-old female infant, the youngest patient. Those 7 cases prior to 2000 were summarized in Table 1.

If the flowchart was presented in a figure, please do not describe in word on the text again. Explain only one time.

à reply: Thank you for your comment. We agree with your comment, and it would be better to make the manuscript concise. In the Materials and Methods, we removed the duplicated sentences, and the corresponding part has been modified as follows. Line 66~68.

Increase the visual appearance of the manuscript within the size of the items such tables and figure. One page could no be longer to have only the flowchart. Please revise.

à reply: Figure 1 was revised using ‘PRISMA_2020_flow_diagram’. Its size was also revised. The size of Figure 2~4 was revised as well according to your comment.

The statistical approach should be explained in a separate section on methodology

à reply: A statistical approach section was added on methodology. Line 76~82.

Risk of bias is mandatory in the systematic review, include on the methodology section and results.

à reply: Risk of bias was presented by adding Figure 5 and 6. Corresponding comments about Risk of bias was inserted in the methodology section (2.4 Quality assessment) and the results (3.8. Quality assessment). Line 83~87 and Line 226~228.

Update the PRISMA checklist to the latest version of 2020 and change the manuscript according to this updated issue.

à reply: PRISMA checklist was updated to the latest version of 2020 and presented in modified Supplementary Table 1.

The results were clearly stated and described but you need to divide into section to improve the readability. It is impossible to read all this text in one time without divisions.

à reply: Thank you for your very good suggestion to improve readability. Results were divided into sections as follows. 3.1. Characteristics of the included cases, 3.2. Demographic and laterality characteristics of the eyelid angiosarcoma 3.3. Overall prognosis of the eyelid angiosarcoma 3.4. Prognosis according to age and laterality of the eyelid angiosarcoma 3.5. Prognosis according to treatment modalities of the eyelid angiosarcoma 3.6. Prognosis according to metastasis of the eyelid angiosarcoma 3.7. Misdiagnosis of the eyelid angiosarcoma 3.8. Presenting features of the eyelid angiosarcoma 3.9. Quality assessment

Improve the visual appearance of the flowchart, use programs such lucid chart to better visuality. Reduce the size. Revise the sum of the articles.

à reply: Thank you for your comments. Figure 1 was revised using ‘PRISMA_2020_flow_diagram’. Its size was also revised. We re-arranged figures and tables to revise the sum of the articles and increase readability.

Try to include table 1 in two pages, When the window time was applied it is probably eased to do this.

à reply: Using additional abbreviations and revising the size of the text, Table 1 is included in two pages.

The discussion was well-written but needs section too.

à reply: Thank you for your comment. Discussion part was divided into sections as follows according to your suggestion. 4.1. Incidence of the eyelid angiosarcoma 4.2. Predominance and metastasis of the eyelid angiosarcoma 4.3. Prognosis of the eyelid angiosarcoma 4.4. Treatments of the eyelid angiosarcoma 4.5. Misdiagnosis of the eyelid angiosarcoma 4.6. Limitations and strengths of this study 4.7. Conclusion and future research

A strengths and limitations section should be added. Furthermore, a future research lines could be added at the end of the discussion

à reply: Strength and limitation section was added. Line 334-342.

Future research was added at the end of the discussion. Line 348-352.

The conclusion could be more soundness in order to give extra power to the research.

à reply: Conclusion was revised concisely for soundness. Line 344-347.

Previous sentence of “Eyelid angiosarcoma occurs more often in old age and is most commonly reported in Caucasian men” was removed. Previous sentence of “Eyelid angiosarcoma is prone to misdiagnosis. If treatments for other misdiagnosed eyelid problems, such as cellulitis or angioedema, are not effective, eyelid angiosarcoma should be suspected and confirmed through repeated biopsies.” was moved to section ‘4.5. Misdiagnosis of the eyelid angiosarcoma’.

We truly appreciate your time and efforts for our study again.

Reviewer 2 Report

In this manuscript the authors review the literature on angiosarcoma of the eyelid and present primarily epidemiologic and treatment outcome results. The authors should include a report of most common presenting features of eyelid angiosarcoma (i.e. isolated mass lesion vs. diffuse eyelid edema, etc), as this is critical to the primary purpose of the paper to increase provider awareness and recognition of this disease entity. A representative clinical image would be useful.

The authors should also expand their discussion of surgical management of this disease, which they indicate is associated with the highest survival, to help guide treatment (i.e. Is surgery performed via wide local excision? Is Mohs appropriate? Frozen intraoperative path? Simultaneous SLNB?), and indicate if one method is particularly correlated with improved survival. 

The authors recommend systemic workup, including PET CT imaging, for patients with this disease process. It would be useful if they were able to report what type of systemic workup was performed pre-treatment in the patients included in the literature they reviewed. 

On lines 96-98, they should refer to the lesion location and laterality by "eyelid", rather than "eye". 

Author Response

Responses to the Reviewers

Reviewer 2

Comments and Suggestions for Authors

In this manuscript the authors review the literature on angiosarcoma of the eyelid and present primarily epidemiologic and treatment outcome results. The authors should include a report of most common presenting features of eyelid angiosarcoma (i.e. isolated mass lesion vs. diffuse eyelid edema, etc), as this is critical to the primary purpose of the paper to increase provider awareness and recognition of this disease entity. A representative clinical image would be useful.

à reply: Thank you very much for the reviewer's very detailed and kind review and comments on this humble study.

Presenting features of the eyelid angiosarcoma were diverse. Looking at the color features first, reddish/ erythematous/rash-like changes were the most common color features in 18 cases, followed by bluish/violaceous/ecchymosis in 8 cases. However, there were 2 cases of yellowish or pale yellow. There were some cases where color features were not mentioned. Shape features of the eyelid angiosarcoma were also presented in various ways. Mass-like lesions such as nodule/papule/plaque were presented in 19 cases. Swelling or edema features were mentioned in 18 cases. In addition, there were cases with macular lesions or only color-changed features. The most common feature of the eyelid angiosarcoma can be briefly summarized as a reddish or bluish maculopapular lesion. However, it is necessary to be careful as there are cases where it appears only with a change of color or only the shape changes without a change of color.

This reply was added in results section (3.8. Presenting features of the eyelid angiosarcoma). Line 212~225.

The authors should also expand their discussion of surgical management of this disease, which they indicate is associated with the highest survival, to help guide treatment (i.e. Is surgery performed via wide local excision? Is Mohs appropriate? Frozen intraoperative path? Simultaneous SLNB?), and indicate if one method is particularly correlated with improved survival. 

à reply: Thank you for your good inquiry.

Of the 42 cases, there were 26 cases in which excision was performed (Table 2). Mohs excision was performed in one case [22], exenteration in two cases [11, 31], anterior orbitotomy in one case [31]. There were 13 cases that clearly stated that wide excision was performed. The treatment modality used in the other 10 cases was mentioned as surgical excision, complete surgical excision, em bloc excision or radical excision. However, most of them seem to have implemented wide excision. In addition, intraoperative frozen margin was confirmed only in 4 cases [22, 27, 33, 36]. Therefore, in most cases, wide excision was performed and the number of surgical cases was only 26, making it difficult to evaluate the difference in prognosis according to the surgical methods.

Although the prognosis of the eyelid angiosarcoma is better than that of angiosarcoma invading the face and scalp, the mortality rate is still high in cases of recurrence or metastasis. Mohs excision can be attempted only when a wide excision is not possible. In addition, it seems necessary to confirm whether the intraoperative frozen margin is negative even at these cases.

This reply was added in discussion section (4.4. Treatments of the eyelid angiosarcoma). Line 308~321.

The authors recommend systemic workup, including PET CT imaging, for patients with this disease process. It would be useful if they were able to report what type of systemic workup was performed pre-treatment in the patients included in the literature they reviewed. 

à reply: Thank you for your comments.

Of the 42 cases, CT was used as a systemic workup in 16 cases [8, 11, 12, 15, 18, 21, 23-26, 29, 31, 34-36], and MRI in 6 cases [9, 11, 18, 21, 26, 31]. Only two cases [26, 30] used PET-CT as a systemic workup.

This reply was added in the result section (3.6. Prognosis of the eyelid angiosarcoma according to metastasis). Line 191~193.

On lines 96-98, they should refer to the lesion location and laterality by "eyelid", rather than "eye". 

à reply: Thank you for your detailed comment. The sentences were modified using “eyelid”. Line 107~110.

We truly appreciate your time and efforts for our study again.  

Round 2

Reviewer 1 Report

comments solved

This manuscript is a resubmission of an earlier submission. The following is a list of the peer review reports and author responses from that submission.

Round 1

Reviewer 1 Report

Dear Authors,

Interesting paper, hovewer in my opinion doubts considering lack of informed consent exist, especially because patient can be identified themself.

Other issues:

Page 13: In exclusion criteria is stated: “4. missing information on demographic characteristics”

And then: the ethnic group could not be identified in 19 cases.

This inconsistency need to be corrected.

Page 14: Are any data of survival time of a single patient, who refused treatment and showed no regression exist?

Is there any explanation of spontaneous regression? (Possible misdiagnosis?)

Page 17-20 – I will indicate both patients, who refused treatment.

Best regards

Reviewer 2 Report

The article is at least confusing, on the one hand is a case report and the Journal Clinical of Medicine do not accept case reports and in the other hand it a literature review when the tendence in the present is to perform systematic review that achieved scientific evidence.

It is very difficult to start the lecture of the work without the presentation requirement of the Journal. It demonstrates that de authors do not read and apply the instruction guidelines of the MDPI web.

The way to express the title is very confusing a do not state clear the topic of the paper.

The title page must be in one page and the abbreviations are not necessary, the authors should present with brackets along the manuscript.

Within the content we recommend not to include the case report and perform a real systematic review with the instruction of PRISMA statement.

In the flowchart diagram appear 3 review comment, please read the review and evaluate if a new review is necessary.

The summary of results must be presented in a different form, you could not present a table of 5 pages.

The structure of a systematics review is different according the PRISMA statement. In the title the authors expressed that a literature review was presented. After a systematic review is writing and finally a narrative review was performed. Please with coherence to the manuscript.

From table 4 to 9 and some figures are not necessary it could resume in a sentence within the most important topic.

In general, the manuscript does not have a manuscript structure and it is like text tables and figure mixed with random order and no sense.

The informed consent to patients is necessary although the faces do not appear on the images.

No minors concerns were identified